# Cytomegalovirus in Adenoma and Carcinoma Lesions: Detecting Mono-Infection and Co-Infection in Salivary Glands

**DOI:** 10.3390/ijms25147502

**Published:** 2024-07-09

**Authors:** Ana Carolina Silva Guimarães, Jéssica Vasques Raposo Vedovi, Camilla Rodrigues de Almeida Ribeiro, Katrini Guidolini Martinelli, Marcelo Pelajo Machado, Pedro Paulo de Abreu Manso, Barbara Cristina Euzebio Pereira Dias de Oliveira, Mariana Lobo Bergamini, Catharina Simioni de Rosa, Tania Regina Tozetto-Mendoza, Ana Carolina Mamana Fernandes de Souza, Marília Trierveiler Martins, Paulo Henrique Braz-Silva, Vanessa Salete de Paula

**Affiliations:** 1Molecular Virology and Parasitology Laboratory, Oswaldo Cruz Institute, Oswaldo Cruz Foundation, 4365 Brasil Ave., Manguinhos, Rio de Janeiro CEP 21040-360, Brazil; anaguimaraes.bio@gmail.com (A.C.S.G.); jessicavasquesr@gmail.com (J.V.R.V.); camilla_almeida@hotmail.com (C.R.d.A.R.); 2Social Medicine Department, Federal University of Espírito Santo, Espirito Santo CEP 2975-910, Brazil; katrigm@gmail.com; 3Pathology Laboratory, Oswaldo Cruz Foundation, 4365 Brasil Ave., Manguinhos, Rio de Janeiro CEP 21040-360, Brazil; mpelajo@gmail.com (M.P.M.); ppmanso@ioc.fiocruz (P.P.d.A.M.); barbara.micro@gmail.com (B.C.E.P.D.d.O.); 4Stomatology Department, Dentistry School, University of São Paulo, São Paulo CEP 05508-000, Brazil; mariana.bergamini@usp.br (M.L.B.); catarinasimioni@usp.br (C.S.d.R.); mariliam@usp.br (M.T.M.); pbraz@usp.br (P.H.B.-S.); 5Virology Laboratory, Tropical Medicine Institute of São Paulo, Medical School, University of São Paulo, São Paulo CEP 05508-000, Brazil; tozetto@usp.br (T.R.T.-M.); anacarolinamamana@yahoo.com.br (A.C.M.F.d.S.)

**Keywords:** HCMV, salivary glands, co-infections

## Abstract

Salivary glands’ neoplasms are hard to diagnose and present a complex etiology. However, several viruses have been detected in these neoplasms, such as HCMV, which can play a role in certain cancers through oncomodulation. The co-infections between HCMV with betaherpesviruses (HHV-6 and HHV-7) and polyomaviruses (JCV and BKV) has been investigated. The aim of the current study is to describe the frequency of HCMV and co-infections in patients presenting neoplastic and non-neoplastic lesions, including in the salivary gland. Multiplex quantitative polymerase chain reaction was used for betaherpesvirus and polyomavirus quantification purposes after DNA extraction. In total, 50.7% of the 67 analyzed samples were mucocele, 40.3% were adenoma pleomorphic, and 8.9% were mucoepidermoid carcinoma. Overall, 20.9% of samples presented triple-infections with HCMV/HHV-6/HHV-7, whereas 9.0% were co-infections with HCMV/HHV-6 and HCMV/HHV-7. The largest number of co-infections was detected in pleomorphic adenoma cases. All samples tested negative for polyomaviruses, such as BKV and JCV. It was possible to conclude that HCMV can be abundant in salivary gland lesions. A high viral load can be useful to help better understand the etiological role played by viruses in these lesions. A lack of JCV and BKV in the samples analyzed herein does not rule out the involvement of these viruses in one or more salivary gland lesion subtypes.

## 1. Introduction

Viruses have been seen as pathogens involved in etiologic factors associated with carcinogenesis. This association was first described by Rous, who showed that sarcomas can be caused by viruses [1,2]. Since then, viral oncogenesis is a field of interest in cancer biology [2]. The development of advanced molecular techniques enabled discovering more viruses and associating them with the cause of neoplasms diagnosed in humans [2]. Several viruses are important co-factors in the development of salivary gland lesions [3]. Among them, one finds human betaherpesviruses, such as Human cytomegalovirus (HCMV), *Human betaherpesvirus* 6A/6B (HHV-6A/6B), *Human betaherpesvirus* 7 (HHV-7), and *Human polyomavirus* (JCV and BKV), which have been detected in saliva samples and salivary gland tissues [4,5,6].

HCMV persists in the salivary gland as a persistent lytic infection and shows tropism towards the salivary gland ductal epithelium, establishing a persistent, lifelong infection after primary exposure [2]. This virus also belongs to the herpesvirus group, which is implicated in breast, brain, lung, colon, prostate, and salivary gland cancer [7,8,9,10,11,12,13]. HCMV infection is often asymptomatic in immunocompetent individuals [14]. Approximately 10% of infected individuals present symptoms, such as self-limiting mononucleosis-like syndrome [14]. HCMV infection in patients with immunodeficiency or hematological disorders presents substantial morbidity and mortality rates in almost all cases, due to uncontrolled viral replication resulting from viral dissemination to multiple organs [13,14,15].

Cytomegalovirus is a highly prevalent herpesvirus worldwide [16]. Its prevalence reaches approximately 100% in both African and Asian countries, as well as 80% in European and North American countries [16,17]. According to epidemiological data provided by Centers for Disease Control and Prevention (CDC) and by the World Health Organization (WHO), HCMV can infect people in all age groups; in the United of States, over 50% of adult individuals are infected with HCMV by the age of 40, whereas one in three children are infected with HCMV by the age of five [16]. There have been a few studies focused on investigating HCMV infection epidemiology in Brazil. Some reports about epidemiological data have shown different HCMV infection prevalence rates in the Santa Catarina (96.4%), Rio de Janeiro (78.7%), and São Paulo (84.8%) states [18,19].

HCMV, HHV-6A/HHV-6B, and HHV-7 belong to family *Herpesviridae* (subfamily *Betaherpesvirinae*); they are highly disseminated worldwide, as well as being associated with several salivary gland lesions [20,21]. Furthermore, these viruses show tropism towards epithelial cells, monocytes, leukocytes, and CD4+ T lymphocytes; therefore, they establish latency in such salivary gland cells [22]. *Human betaherpesvirus* remains latent in host organisms until viral reactivation takes place [22]. In addition, these viruses can be detected in hosts’ saliva throughout their life [23,24].

Among the thirteen polyomaviruses currently classified as belonging to family *Polyomaviridae*, one finds the BKV (BKV) and JCV polyomaviruses (JCV), which are DNA oncogenic viruses often associated with some human cancer types [25]. According to the International Agency for Research on Cancer (IARC), the JCV and BKV viruses are classified as grade 2B carcinogen (likely carcinogenic to humans) [25,26].

BKV and JCV polyomaviruses are endemic in almost all populations in the world; moreover, infections caused by them do not present a well-defined incidence seasonality [27]. Approximately 80–90% of the population are latent hosts for JCV and BKV; primary infections are often subclinical during childhood and they present mild upper respiratory symptoms [25,28]. These viruses can be detected in the urine and saliva samples of asymptomatic healthy adult individuals, and it indicates their tropism towards kidney cells and salivary gland epithelial cells [25]. Some studies have suggested that BKV shows tropism towards oral cavity cells and, recently, they suggested BKV and JCV’s involvement in oral carcinomas [25]. In addition, JCV was detected in human tonsil tissue samples, and this factor was suggested as evidence of infection site [25].

Salivary gland neoplasms are uncommon lesions that widely vary in clinical, histological, and biological aspects [29,30]. Overall, salivary gland lesions are hard to diagnose and differentiate based on morphological aspects [29,30]. Epidemiological data have shown that salivary gland neoplasms account for approximately 3% to 6% of all head and neck tumors—their overall annual incidence reaches 4–13.5 cases per 100,000 individuals [30,31].

The incidence of salivary gland lesions is gradually increasing due to changes in people’s lifestyles and an increasing elderly population [32]. Although the etiology of salivary gland lesions remains unknown, alcohol intake and history of radiotherapy are considered as risk factors for their development [32,33]. In addition to these risk factors, other biological and cellular factors, changes in tumor suppression genes, oncogenes’ activation, and viral infections also work as co-factors for the development of salivary gland lesions [3,32].

In addition to saliva samples, HCMV, among other viruses, such as HHV-6, HHV-7, JCV, and BKV, has been detected in paraffin-embedded samples of salivary gland neoplasms [4,21,26,34,35,36]. The ability to detect viral genomes in salivary gland lesions does not indicate that these viruses can be involved in the development of these lesions, and more studies are necessary to investigate the presence of these viruses and development of these lesions. Based on the literature in this field, there have not been many studies focused on investigating the role played by viral oncogenesis in salivary gland lesions. Therefore, the aim of the current study was to describe HCMV frequency and co-infections in patients presenting neoplastic and non-neoplastic lesions, including in the salivary gland.

## 2. Results

### 2.1. Patients’ Epidemiological and Clinical Features

The patients’ mean age was 31.33 ± 16.48 years; most of them belonged to the female sex (58.2%), at mean age of 33.19 ± 14.6 years, whereas 41.8% of them belonged to the male sex, at mean age of 29.58 ± 18.27 years. Lesions were classified as neoplastic or non-neoplastic. In total, 34 (34; 50.7%) of the 67 analyzed patients had non-neoplastic lesions of the mucocele type, 27 (40.3%) presented neoplastic lesions of the pleomorphic adenoma type, and 6 (8.9%) presented neoplastic lesions of the mucoepidermoid carcinoma type.

With respect to sex-wise distribution, the mucocele lesion type was more prevalent in individuals belonging to the male sex (55.9%) than in those belonging to the female sex. On the other hand, the adenoma and carcinoma lesion types were more prevalent in individuals belonging to the female sex than in those belonging to the male sex; this difference was statistically significant (*p* = 0.045)

Salivary gland lesions were prevalent in all ages between 0 and ≥ 40 years. Lesion severity increased as individuals got older. Individuals in the age groups of 0–19 and 20–39 years mostly presented the mucocele and adenoma lesion types, whereas individuals older than 40 years mostly presented neoplastic lesions of the carcinoma type (80.0%) (Table 1).

### 2.2. Detecting HCMV, HHV-6 and HHV-7 in Salivary Gland Lesions

Only 18 (26.86%) of the 67 processed samples tested negative for all analyzed viruses, 10 (55.55%) were mucocele, 7 (38.88%) were adenoma, and 1 (5.55%) was carcinoma. HCMV recorded the highest viral load (1.69 × 10^12^) in comparison to HHV-6 (7.79 × 10^11^) and HHV-7 (8.80 × 10^11^). HHV-6 recorded the highest viral load in the carcinoma lesion type, whereas HCMV and HHV-7 prevailed in the adenoma and carcinoma lesion types, respectively. A Tukey test showed a statistically significant difference in HHV-6 viral load between the mucocele and carcinoma lesion types, and between the adenoma and carcinoma types (*p* ≤ 0.001) Figure 1.

### 2.3. Presence and Absence of HCMV, HHV-6 and HHV-7 in Salivary Gland Lesions

Most samples tested positive for HCMV (52.2%). HCMV prevailed in the adenoma and carcinoma lesions, although HHV-6 prevailed in the mucocele lesion type. However, there was no statistically significant difference between HCMV, HHV-6, and HHV-7 and lesion types (Table 2).

### 2.4. Detecting BKV and JCV Polyomaviruses

All the analyzed samples tested negative for JCV and BKV polyomaviruses.

### 2.5. Detecting HCMV in Patients

Most patients presented infection caused by HCMV. HCMV detection was more prevalent in individuals belonging to the female sex than in those belonging to the male sex—66.7% and 33.3%, respectively. HCMV DNA detection among individuals in the age groups of 0–19 and 20–39 years reached 40%, whereas only 20% of individuals in the age group of ≥40 years presented HCMV infection. However, there was no statically significant difference in HCMV infection between patients’ sex and age (0.268).

### 2.6. HCMV Co-Infection and Mono-Infection in Salivary Lesion Types

HCMV mono-infection and adenoma lesions were the most prevalent types observed in the assessed patients—13.4% and 25.9%, respectively. Twenty-six (26) samples presented co-infections. There was high rate of triple infection with HCMV/HHV-6/HHV-7 (20.9%), whereas co-infection with HCMV/HHV-6 and HCMV/HHV-7 reached 9.0%, for both infections. Triple infection (HCMV/HHV-6/HHV-7) and co-infections (HCMV/HHV-6 and HCMV/HHV-7) recorded high rates in patients with the mucocele lesion type (Table 3).

Spearman correlation between the HCMV/HHV-7 (rho-spearman = 0.465; *p*-value = 0.003) and HHV-6/HHV-7 (rho-spearman = 0.425; *p*-value = 0.012) viral loads showed that the HHV-7 viral load increased as the HCMV viral load also increased. Moreover, the HHV-7 viral load increased as the HHV-6 viral load also increased.

## 3. Discussion

Studies focusing on investigating HCMV, HHV-6 and HHV-7 co-infection, and HCMV mono-infection in salivary gland lesions remain scarce in the literature. The aim of the current study was to describe HCMV frequency and co-infections in patients presenting neoplastic and non-neoplastic lesions in salivary glands. HCMV was herein detected in most patients. This finding can be explained by HCMV tropism towards salivary glands’ epithelial cells, through viral detection in these organs [23,37].

A study previously carried out by our research group evidenced that salivary glands play important role as site for active and persistent infections caused by betaherpesvirus, as well as that high viral loads are correlated with mRNA detection levels, which, in turn, suggests active HHV-6 replication [24,38].

HCMV infection leads to the progression/development of several cancer types in humans; this factor is associated with the capacity of the virus to contribute to carcinogenesis as an initiator or promoter [39,40,41]. HCMV allows the tumor to escape immune surveillance by encoding viral proteins and inducing various immunosuppressor cellular mechanisms, which stimulate tumor growth [41]. This finding suggested HCMV addition to the group of viruses capable of inducing malignant transformation [41]. The role of HCMV oncomodulation has been implicated in promoting tumor growth by affecting various cellular processes, including inflammation, immune evasion, and cellular proliferation [41,42].

Oncogenic DNA virus had been previously reported in association with salivary gland tumors; HCMV DNA was detected in five salivary gland tissue samples collected from patients with salivary gland neoplasms, as well as in one healthy salivary gland tissue sample [43]. These findings have shown that saliva can play important role in HCMV transmission. Moreover, the fact that this virus was found in saliva samples indicates that it has viral activity in salivary glands. In our study, a statically difference between virus and lesion types was not found, 50.7% of patients had HCMV associated with non-neoplastic lesions and 49.2% had HCMV associated with neoplastic lesions. However, the exact mechanisms through which HCMV might contribute to lesion of salivary glands are not fully understood, and the relationship between them is still an area that needs active research. There is evidence suggesting that HCMV might play a role in certain cancers through oncomodulation rather than direct oncogenicity [41,42,44].

Triple infection with HCMV, HHV-6, and HHV-7 was detected in 20.9% of the samples, which recorded the highest detection rates. Just like HCMV, HHV-6 and HHV-7 also replicate in salivary gland cells, and may be detected in these organs [23,37]. These viruses can cause infections during infancy and become latent for the rest of hosts’ lives [45,46]. A study conducted with 182 individuals in the age group of 14–56 years evidenced HHV-6 and HHV-7 prevalence in 9.8% and 12.6% of the assessed population, respectively [47]. Studies conducted on individuals with history of transplant surgery have shown that HHV-6 reactivation is caused by co-infection with HHV-7, and that HCMV infection is often associated with HHV-6 and HHV-7 viral reactivation [24,47].

Mucocele was the most common lesion type among the patients assessed herein. Mucocele and pleomorphic adenoma are the lesion types most often observed among patients, and this corroborates the current findings [48,49]. A study conducted in Brazil, which used 88,430 oral and maxillofacial lesions, showed that 2292 (2.6%) of samples presented salivary gland lesions. With respect to benign tumors, 1086 (47.4%) samples were classified as pleomorphic adenomas, whereas 322 (14.0%) were classified as mucoepidermoid carcinomas, which is the most common malignant tumor found among patients with salivary gland lesions [49].

In addition to the sex-related influence observed herein, the patients’ ages were also correlated with the salivary gland lesion type. Patients in the age group of 20–39 years and those over 40 years old recorded a high prevalence of pleomorphic adenoma lesions—36.4% and 37.0%, respectively. However, individuals over 40 years old also recorded a prevalence of mucoepidermoid carcinoma (80.0%). Epidemiological data have shown that pleomorphic adenoma lesions can emerge at any age, although they prevail in adult individuals in the age group of 40–60 years [50,51,52,53,54]. Mucoepidermoid carcinomas have previously been reported in individuals in the age group of 16–64 years, whereas pleomorphic adenomas have been reported in individuals in the age group of 14–95 years [55]. These findings evidence that salivary gland lesions can happen in both old and young individuals.

Cytomegalovirus and HHV-6 were detected in patients presenting the mucocele and adenoma lesion types. Some studies have shown that *Human betahervespiruses* can transactivate each other, whereas HCMV infection triggers HHV-6 and/or HHV-7 co-infection [46,56,57,58,59,60]. HCMV and HHV-6 incidence in certain neoplasm types has been described in the literature; however, the detection of these viruses in neoplastic cells is insufficient for defining the direct role played by them in carcinogenesis [61]. In addition, HHV-6 is not seen as directly oncogenic and it can act as a cofactor capable of indirectly stimulating tumor cell growth; in some cases, it synergistically acts along with other viruses [61].

Although HHV-6 and HHV-7 were detected in the patients and prevailed in certain salivary gland lesion types, the data analyzed in the current study did not show a statistically significant association between the incidence of these viruses and pathogenesis interaction. Similar data were observed in a study conducted in Iran with paraffin-embedded salivary gland samples. HHV-6 was detected in 6.6% of pleomorphic adenoma samples; HHV-6/HHV-7 co-infection accounted for 13.3% of samples, although there was no significant association between virus and lesion types [36].

Triple infection with HCMV, HHV-6, and HHV-7 was most often observed in salivary gland lesions. Co-infection with HCMV, HHV-6, and HHV-7 was observed in a study conducted with patients with leukemia, who presented IgG and IgM antibodies. The results showed that 15.8% of them were co-infected with HCMV/HHV-6, 5.3% were co-infected with HCMV/HHV-7, and 2.1% were co-infected with HCMV/HHV-7 [46]. These findings evidence that these viruses can co-exist.

Co-infection with HCMV and polyomavirus was also investigated in the current study. However, all 68 paraffin-embedded samples tested negative for both JCV and BKV. Although the analyzed data evidenced a lack of these viruses in salivary gland lesions, studies available in the literature have reported these viruses in saliva samples and salivary gland tissues of immunocompromised, transplanted, and healthy patients. This indicates that these viruses show tropism towards salivary glands [25,26,32,34,62,63,64].

According to the literature in this field, JCV and BKV viruses have been detected in salivary gland tissues [43]. JCV and BKV incidence was assessed in a study conducted in the United States (US) with paraffin-embedded samples collected from patients with malignant and non-malignant cancer types. The aforementioned viruses were the most common pathogens found in neoplastic (36.6%) and healthy salivary gland samples (39.2%); they also recorded a high detection rate in non-malignant tissues (60%) [43]. Hametoja and colleagues showed that only JCV DNA—among the JCV, BKV, and SV40 polyomaviruses—was found in adenoid cystic carcinoma samples—10.3% of 68 samples tested negative for the other polyomaviruses assessed in their study [25].

JCV and BKV viruses have been found in healthy individuals, as well as detected in saliva samples deriving from HIV-positive and kidney transplant patients [63,64,65,66,67,68,69]. All the patients in the current study were tested for HIV and recorded negative results. In addition, none of them had undergone a kidney transplant, and this may be one of the reasons why these samples tested negative for these viruses. Although these individuals tested negative for JCV and BKV viruses, this finding does not mean that these individuals were infection-free. Besides these factors, some points can be taken into consideration when it comes to the absence of these viruses, namely: primary infection with BKV and JCV viruses, viral latency in other organs like kidneys, or lack of infection caused by these viruses. Other sample types must be tested and analyzed in order to accurately prove the presence of these polyomaviruses.

Although DNA from tissue samples stored in paraffin blocks enables conducting retrospective studies, the current study has some limitations, namely: the process of cutting formalin-fixed tissues can lead to false negative results, depending on the region where the cut was made, or even to regions with a lower or higher viral load. Furthermore, it was not possible to compare these samples to samples deriving from salivary glands without lesions.

Viral HCMV DNA was observed herein in both neoplastic and non-neoplastic lesions found in salivary glands; HCMV recorded the highest viral load in comparison to HHV-6 and HHV-7. Although HCMV is not often seen as an oncogenic virus, viral DNA, mRNA, and antigens’ detection in tumor tissues has suggested the role played by HCMV infection in the etiology of human malignancies. HCMV induces oncomodulation and leads to proteins and noncoding RNA activation, which, in turn, can influence tumor cell properties, such as cell proliferation and invasion [70]. Moreover, co-infections between HCMV and other betaherpesvirus types, such as HHV-6 and HHV-7, were observed herein. According to studies available in the literature, these co-infections can contribute to the malignancy of certain cancer types, since these viruses can have a synergistic effect on viral replication.

The results recorded herein for HCMV detected in salivary gland lesion regions confirm that HCMV infects the salivary glands, and that saliva is an important transmission route used by it. Furthermore, HCMV co-infection with HHV-6 and HHV-7 in the investigated neoplasms suggests that the synergy of these viruses can induce viral replication and increase viral load. However, further studies should be carried out to establish the association between these viruses and whether their synergy can over-activate some oncogenic regulatory pathways and epidermal growth factors.

## 4. Materials and Methods

### 4.1. Study Population

In this study, the samples collected were lesions, including those in the salivary gland. All the samples had lesion cells. The samples were obtained at the archives of the Surgical Oral and Maxillofacial Pathology Service of the Stomatology Department of the Dentistry School, University of São Paulo, Brazil. This study was approved by the Research Committee of the Dentistry School, University of São Paulo (approval number: 3683851).

This retrospective study was performed from 2011 to 2019, and it included patients in the age group of 5 months–68 years, who presented some of the most common salivary lesion types, such as pleomorphic adenomas, mucoceles (benign lesions), and mucoepidermoid carcinomas (malignant lesions). These lesions were located on the patients’ tongue, labial mucosa, lip, hard palate, and floor of the mouth. The inclusion criteria comprised the incidence of salivary gland lesions, as well as the quality of the paraffin tissue and histological sections. Samples that did not have enough material for all the subsequent molecular analyses were excluded from the study.

Samples were classified according to diagnostic criteria set by the World Health Organization for classification purposes. Mono-infection and co-infection with cytomegalovirus were analyzed.

### 4.2. Nucleic Acids’ Extraction

Nucleic acids were isolated from 4µm of tissue, based on using the MagMax FFPE DNA/RNA Ultra Kit (Thermo Fisher Scientific, Waltham, MA, USA). The extracted samples were stored at −70 °C until the analysis time.

### 4.3. Multiplex Real-Time Quantitative Polymerase Chain Reaction (qPCR) Analysis Applied to HCMV, HHV-6, and HHV-7

Quantitative/Real-time (qPCR) was performed, based on using commercial TaqMan™ Universal PCR Master Mix (Thermo Fisher Scientific, Waltham, MA, USA), in order to confirm viral detection, as well as to measure the viral load through the HCMV, HHV-6, and HHV-7 target regions U54, U56, and U37, respectively. However, this assay could not differentiate HHV-6A from HHV-6B in the analyzed samples. Multiplex qPCR was performed according to the manufacturer’s instructions; a reaction mixture comprising 1 μL of 25× PCR Enzyme, 1 μL of each oligonucleotide (3 μM), 1 μL of each probe (0.4 μM), 12.5 μL of 1× PCR Buffer, and 2.5 μL DNA.

Oligonucleotides, probes, and synthetic standard curves used herein for betaherpesvirus detection purposes had been previously described by Raposo et al. [71]. Synthetic standard curves ranging from 5 to 5 × 10^8^ copies/μL were used for absolute viral DNA quantification. Ultrapure water and negative samples were used as negative controls and positive samples were used as positive controls.

### 4.4. Multiplex Real-Time Quantitative Polymerase Chain Reaction (qPCR) Analysis Applied to Human Polyomavirus (JCV and BKV) Performed using QuantStudio 3

Polyomaviruses were detected and quantified based on the method described by Castro and colleagues [72]. Oligonucleotide probes used to detect polyomaviruses target highly conserved regions in the viral genome of published strains. Ag-T regions were used herein for the amplification of both BKV and JCV viruses. This assay enables differentiating BKV from JCV. Multiplex qPCR was performed according to the manufacturer’s instructions using TaqMan™ Universal PCR Master Mix (Thermo Fisher Scientific, Waltham, MA, USA); a reaction mixture comprising 1 μL of 25× PCR Enzyme, 1 μL of each oligonucleotide (3 μM), 1 μL of each probe (0.4 μM), 12.5 μL of 1× PCR Buffer, 3 μL of water DNA/RNAse free, and 2.5 μL DNA.

Oligonucleotides and probes used herein to detect polyomaviruses were described by Castro and colleagues [72]. Ultrapure water and negative samples were used as negative controls, whereas previously positive samples were used as positive controls.

### 4.5. Synthetic Standard Curves Plotted to Quantify JCV and BKV Polyomaviruses and HCMV, HHV-6, and HHV-7 Betaherpesviruses

A synthetic curve is a Ultramer DNA Oligos hybridized with a probe and two primers. Ultramer DNA Oligonucleotides are generated through proprietary synthesis methods capable of delivering high-quality oligos up to 200 bases (Integrated DNA Technologies, Inc., Coralville, IA, USA).

Synthetic standard curves ranging from 5 to 5 × 10^10^ copies/µL were plotted for the herein conducted absolute viral DNA quantification. The adopted detection limit was 5 copies/mL. JCV (5′TTCGTGGAAAGTCTTTAGGGTCTTCTACCTTTCGTATTCTTTTTAGGTGGGGTAGAGTGTTGGGATCCTATGCGTTTTTCATCATCACTGGCAAACATCTGATA3′) and BKV (5′TTCGTGGCTGAAGTATCTGAGACTTGGGCGTATGAGCATTGTGATTGGGATTCAGTGCTTGATGCGTTCCATGTCCAGAGTCTTCAGTTTCCTGATA 3′) were the herein used sequences. Synthetic standard curves for betaherpesvirus, (HCMV 5′-TTCGTGGCCTCGTAGTGAAAATTAATGGTCGTATTTGAACAGATCGCGCACCAATACGGATGCGTTCCTGCAGACAGTAACGGCCCTGATA-3′,HHV-6 5′-TTCGTGCAAGCTCATGAACATCGTCACGTATACCGATCCCAGCTCACCACCATCTAAATGCGTAGGTAGCGGCAATTTAGGTCTTTCTGATA-3′ and HHV-7 5′-TTCGTCCAATCCTTCCGAAACCGATCGTATCATGGCCAACAAGCAATCTGCGAGATGCGTTTGTCATTACTCCAGTGA CTTCCGCTGATA-3′) ranging from 5 to 5 × 10^8^ genome copies/µL of HCMV, HHV-6 and HHV-7, the synthetic curves were previously described by Raposo and Colleagues [71].

### 4.6. Statical Analyses

Descriptive statistics of the qualitative variables were determined based on frequency distribution, whereas quantitative variables were determined based on mean values (SD). A chi-square test at 95% CI and *p*-value ≤ 0.05 was used to compare rates between groups (mucoceles, adenomas, and carcinoma lesions; and HCMV infection or no infection). An ANOVA test at 95% CI and *p*-value ≤ 0.05 was performed to assess the difference in the mean (SD) values recorded for viral load based on lesion type. Tukey test was used to identify the lesions presenting differences in viral load for each virus. Rho Spearman was performed to assess the correlation among the HCMV, HHV-6, and HHV-7 viral loads at 95% CI and *p*-value ≤ 0.05. Statistical analyses were performed in Statistical Package for Social Sciences (SPSS) software, version 19.0 (IBM Corp., Armonk, NY, USA).

## 5. Conclusions

The current study determined HCMV frequency and co-infections among HCMV, HHV-6, and HHV-7 in salivary glands presenting neoplastic and non-neoplastic lesions. A high mono-infection with HCMV rate was detected herein in patients with the adenoma lesion type, whereas triple infection was detected in patients with the mucocele lesion type. However, it was not possible to establish a causal relationship between HCMV detection and salivary gland lesions.

Correlation was only observed between HHV-6 viral load and the mucocele/carcinoma lesion types. Although this virus was found herein in neoplastic tissue, it is not seen as directly oncogenic; however, it can be found in some cancer types.

HCMV mono-infection and co-infection recorded high incidence rates in the individuals assessed herein. This finding shows that HCMV can be detected in both neoplastic and non-neoplastic salivary gland lesions and that it may play important role in further studies about the molecular epidemiology of both this virus and cancer type.

A lack of polyomaviruses in paraffin-embedded samples does not rule out any involvement of human JCV and BKV polyomavirus in one or more salivary gland lesion subtypes. The high HCMV, HHV-6, and HHV-7 viral loads in salivary glands may be essential to help in better understanding the association between human betaherpesviruses and the pathogenesis of salivary gland lesions. Although the presence of HCMV, HHV-6, and HHV-7 does not indicate that it plays oncogenic role, the data analyzed herein provide important information to be used in future investigations.

## Figures and Tables

**Figure 1 ijms-25-07502-f001:**
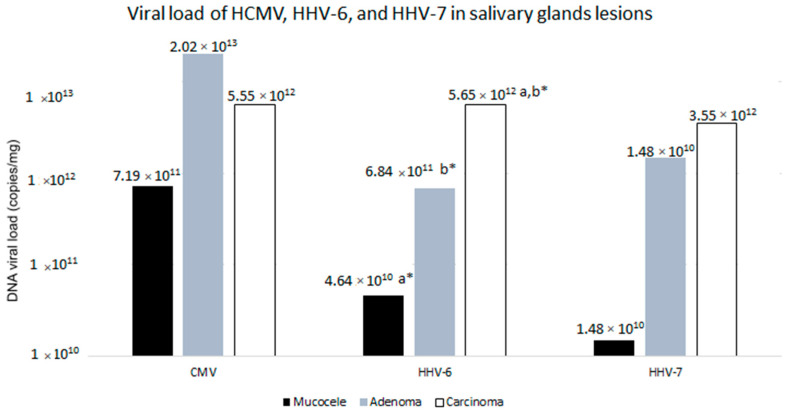
*^,a,b^ Tukey Test performed to analyze the difference between viral load and lesion type. Anova/Tukey showed statistical significance in HHV-6 viral load between the mucocele and carcinoma lesion types and between the adenoma and carcinoma types (*p* ≤ 0.001).

**Table 1 ijms-25-07502-t001:** Association between patients and salivary gland lesions.

	Total	Mucocele	Adenoma	Carcinoma	χ² *p*-Value
Variable	n (%)	n (%)	n (%)	n (%)	
Sex					0.045
Female	39 (58.2)	15 (44.1)	19 (70.4)	05 (83.3)	
Male	28 (41.8)	19 (55.9)	08 (29.6)	1 (16.7)	
Age (years)					<0.001
0–19	20 (30.8)	18 (54.5)	02 (7.4)	0 (0)	
20–39	28 (43.1)	12 (36.4)	15 (55.6)	01 (20.0)	
≥40	17 (26.2)	03 (9.1)	10 (37.0)	04 (80.0)	
Total	67 (100)	34 (50.7)	27 (40.3)	06 (8.9)	

**Table 2 ijms-25-07502-t002:** Association between viruses and lesion types.

	Total	Mucocele	Adenoma	Carcinoma	χ² *p*-Value
Variable	n (%)	n (%)	n (%)	n (%)	
HCMV					0.611
Yes	35 (52.2)	16 (47.1)	15 (55.6)	04 (66.7)	
No	32 (47.8)	18 (52.9)	12 (44.4)	02 (33.3)	
HHV-6					0.273
Yes	31 (46.3)	19 (55.9)	10 (37.0)	02 (33.3)	
No	36 (53.7)	15 (44.1)	17 (63.0)	04 (66.7)	
HHV-7					0.837
Yes	26 (38.8)	13 (38.2)	10 (37.0)	03 (50.0)	
No	41 (61.2)	21 (61.8)	17 (63.0)	03 (50.0)	

**Table 3 ijms-25-07502-t003:** HCMV co-infections and mono-infections in salivary gland lesions.

	Total	Mucocele	Adenoma	Carcinoma
	N (%)	34	27	06
HCMV	9 (13.4)	1 (2.9)	7 (25.9)	1 (16.7)
HHV-6	8 (11.9)	5 (14.7)	2 (7.4)	1 (16.7)
HHV-7	3 (4.5)	2 (5.9)	1 (3.7)	0 (0)
HCMV/HHV-6	6 (9.0)	5 (14.7)	1 (3.7)	0 (0)
HCMV/HHV-7	6 (9.0)	2 (5.9)	2 (7.4)	2 (33.3)
HCMV/HHV 6/HHV-7	14 (20.9)	08 (23.5)	5 (18.5)	1 (16.7)
No infection	21 (31.3)	11 (32.4)	09 (33.3)	1 (16.7)

## Data Availability

Data is contained within the article.

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
