# Peer review of "Cytomegalovirus in Adenoma and Carcinoma Lesions: Detecting Mono-Infection and Co-Infection in Salivary Glands"

_ijms, 2024, doi:10.3390/ijms25147502_

Round 1
Reviewer 1 Report (New Reviewer)
Comments and Suggestions for Authors
The manuscript by Silva Guimarães et al catalogues the detection of HCMV and other members of the betaherpesvirus sub-family as well as polyomaviruses in a salivary gland lesion patient population. It is, though, unclear to this reviewer what the authors actually think this means. Their introduction says (lines 110-111) says that “the aim of the current study was to describe HCMV frequency and co-infections in patients presenting neoplastic and non-neoplastic lesions in salivary glands”. The study certainly does do this but there is, for this reviewer, no scientific application or advancement of our understanding of the effects of such co-infections - which the authors need to experimentally address.
In detail:
1) Introduction.
(i) The authors conflate data on MCMV infection of the mouse and HCMV infection of the human – there is to my knowledge, unlike MCMV, no evidence that HCMV maintains a latent infection in the salivary gland (line 50); it certainly persists in the salivary gland but likely as a persistent lytic infection (e.g. in toddlers).
(ii) Line 105-107. The ability to detect viral genomes in salivary gland lesions does not “indicate that these viruses can be involved in the development of these lesions”.
2) Results.
(i) I note that 50.7 % of patients had HCMV associated with non-neoplastic lesions and 49.2% had HCMV associated with neoplastic lesions and line 154 clearly states that “there was no statistically significant difference between HCMV, HHV-6 and HHV-7, and lesion types” – this hardly supports later suppositions that HCMV may be “oncogenic”.
(ii) Figure 1. Th authors state that Only 18 of the 67 processed samples tested negative for all analyzed viruses – what type of lesions did these patients have? Was the pathology worse in patients in which lesions were associated with viruses?
(iii) Line 178-179. I may have missed this but where are the data that show that “HHV-7 viral load increased as the HCMV viral load also increased”?
3) Discussion
(i) Line 198-199. Neither of the quoted papers (reference 24 and 35), to my knowledge, analyses or shows latency of HCMV in salivary glands.
(ii) Line 204-208. The view that HCMV is an oncogenic virus is extremely contentious (most laboratories believe that, in general, HCMV clinical isolates are, in themselves, not capable of transforming cells) and my view is that the consensus actually is that e.g. HCMV IE1 is not an oncogene.
Similarly, the reference (from 1973….) used by the authors to support the view that HCMV can induce malignant transformation (reference 43) is unsupported by multiple, and more recent, analyses.
In essence, the authors need to balance the arguments for and against HCMV being an oncogenic virus,
(iii) Line 218. As before, reference 24 and 38 used by the authors do not address HCMV latency in the salivary gland – they detect infectious virus.
(iv) Line 301. The authors appear to lack consistency in their key views as they now believe that “HCMV is not often seen as oncogenic virus”. Indeed, I would suggest that the consensus view of the field is more aligned to the authors’ statement in lines 303-305 which is that HCMV induces oncomodulation which can influence tumor cell properties, such as cell proliferation and invasion.
Again, the authors need to balance the arguments for and against HCMV being an oncogenic virus.
(v) Line 312-213. Again, where are the data that say “HCMV co-infection with HHV-6 and HHV-7 in the investigated neoplasms has suggested that the synergy of these viruses can induce viral replication and increase viral load” – particularly that it can induce viral replication and increase viral load”?
Minor comments
(i) Line 208 – addiction?
Author Response
Please see the attachment.

Reviewer 2 Report (New Reviewer)
Comments and Suggestions for Authors
The article describe the frequency of HCMV and co-infections in patients presenting neoplastic and non-neoplastic in lesion including the salivary gland. The authors cover all aspects with details
The results are useful to better understanding the etiological role played by viruses in salivary gland oncogenesis. Furthermore,
1. the authors focused on hard to diagnostic Salivary glands’ neoplasms
2. ''there are not many studies focused on investigating the role played by viral oncogenesis in salivary gland lesions''
3. ''This retrospective study was performed from 2011 to 2019, and it included patients in the age group 5 months-68 years''
The authors fully described:
1. the difference between viral load and lesion
2. association between viruses and lesion types
3. HCMV co-infection and mono-infection in salivary lesion types
Discussion section is exhaustive.
The article is valuable information about hard to diagnostic Salivary glands’ neoplasms
I would like to suggest an improvement for review:
line 355 equipment should be added
line 359 and line 368 references should be added (Castro and...)
Round 2
Reviewer 1 Report (New Reviewer)
Comments and Suggestions for Authors
The manuscript by Guimarães et al is a revised version of a paper I previously reviewed and for which I requested additional information and clarifications.
The authors have made efforts to address many of my comments mainly by textual changes.
In essence, they have toned down much of their original text on the location of HCMV latency as well as statements pertaining to the view that HCMV is an oncogenic virus.
That said, lines 21-23 still states “..several oncogenic viruses have been detected in these neoplasms, such as HCMV…”. The authors need to also modify this statement as they have done throughout the rest of the manuscript.
Similarly, lines 229-230 “Thus, HCMV can be associated with salivary gland neoplasms due to its malignant transformation ability” is not supported by the data presented as HCMV was present in more non-neoplastic lesions and there are no data to support the view that HCMV caused the malignant transformation when it was detected.
Additionally:
1) The revised text (line 52-54) stating “Although some authors have been suggested that MCMV latency can be stablished in salivary glands [7]. Until now, there is few evidence. It is knowing that HCMV persists in the salivary gland as a persistent lytic infection.” Makes little sense. Are they trying to say that MCMV latency can be established in the salivary gland in the mouse but, in humans, HCMV is believed to establish a persistent productive infection in the salivary gland?
Comments on the Quality of English LanguageModerate editing of English language required
Author Response
Please see the attachment

This manuscript is a resubmission of an earlier submission. The following is a list of the peer review reports and author responses from that submission.
Round 1
Reviewer 1 Report
Comments and Suggestions for Authors
In this manuscript, Guimaraes et al. describe a study focused on determining the abundance of several tumor-associated viruses in salivary gland tumors collected from 2011-2019 in Brazil. They extract DNA from paraffin-embedded formalin fixed tissue and screen the DNA by quantitative PCR for the presence of cytomegalovirus, human herpesvirus 6, human herpesvirus 7, and two polyomaviruses, JCV and BKV. They find that the herpesviruses are associated with multiple tumor types, often as a co-infection with multiple viruses detected. They did not detect the polyomaviruses in any of the tissues that they assayed. These findings build support for previously published work showing that CMV is associated with various tumors, including salivary gland tumors. They will be of interest to those focused on understanding the role of these viruses in various tumors.
I have listed below several major problems with the study that limit the interpretation of the authors’ findings. I have also listed several minor issues that might clarify small portions of the manuscript.
Major Problems:
1. CMV is not currently considered an oncogenic virus. It has been found to be associated with tumors, but it is not clear whether it replicates in tumor tissue or is the cause of the tumor (oncomodulation vs. oncogenesis), as the authors correctly describe in Lines 281-285 and 381-383. To support a claim otherwise, the authors should provide direct evidence in the study or citations from the literature. This issue should be addressed in:
a. Lines 21
b. Lines 197-199
2. Viral loads in Table 2 and lines 137-138 should be normalized to number of cells analyzed (e.g., detection of a housekeeping gene or weight of tissue processed)
3. In the absence of negative control tissues for quantitative PCR analysis, standard curves should be presented for each primer/probe set.
4. It is not clear what the authors are stating in Lines 167-172 and Table 4. Are the authors simply showing the percentage of people in their study group that were infected with HCMV? “Mono-infection or co-infection” is a bit confusing. Maybe just state HCMV infected?
5. Again, in lines 194-195, it might be clearer to simply state HCMV infection.
Minor Issues:
1. Line 45 is missing a parenthesis following the abbreviation “HCMV”
2. Line 73: Formatting of the Citation should be addressed
3. Line 120, 140, 149: spelling of “statical” should be corrected to “statistical”
4. Line 140 and footnote for Table 2: did the authors perform a “Tukey” test, not a “Turkey” test?

Comments on the Quality of English LanguageOverall, the quality of English Language was very good. I found a few minor issues, described in my suggestions to authors, that should be addressed to clarify the manuscript.
Author Response
In this manuscript, Guimaraes et al. describe a study focused on determining the abundance of several tumor-associated viruses in salivary gland tumors collected from 2011-2019 in Brazil. They extract DNA from paraffin-embedded formalin fixed tissue and screen the DNA by quantitative PCR for the presence of cytomegalovirus, human herpesvirus 6, human herpesvirus 7, and two polyomaviruses, JCV and BKV. They find that the herpesviruses are associated with multiple tumor types, often as a co-infection with multiple viruses detected. They did not detect the polyomaviruses in any of the tissues that they assayed. These findings build support for previously published work showing that CMV is associated with various tumors, including salivary gland tumors. They will be of interest to those focused on understanding the role of these viruses in various tumors.
I have listed below several major problems with the study that limit the interpretation of the authors’ findings. I have also listed several minor issues that might clarify small portions of the manuscript.
We are grateful for the compliments given to the manuscript that will improve the final version.
Major problems:
- CMV is not currently considered an oncogenic virus. It has been found to be associated with tumors, but it is not clear whether it replicates in tumor tissue or is the cause of the tumor (oncomodulation vs. oncogenesis), as the authors correctly describe in Lines 281-285 and 381-383. To support a claim otherwise, the authors should provide direct evidence in the study or citations from the literature. This issue should be addressed in:
- Lines 21
- Lines 197-199
As suggested, this information was addressed in these lines and other studies were included.
L21: However, several oncogenic viruses have been detected in these neoplasms, such as HCMV, which has the ability to induce malignant transformation and some co-infections between betaherpesviruses (HHV-6 and HHV-7) and polyomaviruses (JCPyV and BKPyV).
L206-211: HCMV infection leads to the progression/development of several cancer types in humans; this factor is associated with the presence of several oncogenes, such as IE1, IE2, US28 and UL76, in its genome [40]–[42]. A study conducted in 1973 has evidenced the oncogenic potential of HCMV, which was capable of changing hamster embryo fibroblast cells. This finding suggested HCMV addiction to the group of viruses capable of inducing malignant transformation [43].
And a conclusion about the oncogenesis was addressed in L216-217: Thus, HCMV can be associated with salivary gland neoplasms due to its malignant transformation ability.
- Yu, S. He, W. Zhu, P. Ru, X. Ge, and K. Govindasamy, “Human cytomegalovirus in cancer: the mechanism of HCMV-induced carcinogenesis and its therapeutic potential,” Front Cell Infect Microbiol, vol. 13, p. 1202138, Jun. 2023, doi: 10.3389/FCIMB.2023.1202138/BIBTEX.
- Shen, H. Zhu, and T. Shenk, “Human cytomegalovirus IE1 and IE2 proteins are mutagenic and mediate ‘hit-and-run’ oncogenic transformation in cooperation with the adenovirus E1A proteins,” Proceedings of the National Academy of Sciences, vol. 94, no. 7, pp. 3341–3345, Apr. 1997, doi: 10.1073/PNAS.94.7.3341.
- Maussang et al., “Human cytomegalovirus-encoded chemokine receptor US28 promotes tumorigenesis,” Proc Natl Acad Sci U S A, vol. 103, no. 35, pp. 13068–13073, Aug. 2006, doi: 10.1073/PNAS.0604433103/SUPPL_FILE/04433FIG6.JPG.
- K. Siew, C. Y. Duh, and S. K. Wang, “Human cytomegalovirus UL76 induces chromosome aberrations.,” J Biomed Sci, vol. 16, no. 1, p. 107, Nov. 2009, doi: 10.1186/1423-0127-16-107/FIGURES/6.
- Albrecht and F. Rapp, “Malignant transformation of hamster embryo fibroblasts following exposure to ultraviolet-irradiated human cytomegalovirus,” Virology, vol. 55, no. 1, pp. 53–61, Sep. 1973, doi: 10.1016/S0042-6822(73)81007-4.
Viral loads in Table 2 and lines 137-138 should be normalized to number of cells analyzed (e.g., detection of a housekeeping gene or weight of tissue processed)
Unfortunately, a normalizer was not inserted, as absolute quantification was performed. Relative quantification was not performed, as it was a cross-sectional study. Other authors who did similar work also did not use normalizers. All samples were measured before extraction, reaching about 4 µm.
In the absence of negative control tissues for quantitative PCR analysis, standard curves should be presented for each primer/probe set.
Synthetic standard curves were used for viral load quantification. This information was addressed in section and the references were add in the manuscript.
Synthetic standard oligonucleotides curves ranging from 5 to 5 x1010 copies/µL were designed for this study for the absolute quantification of viral DNA. The limit of detection was 5 copies/mL. The sequences are JCPyV (5´TTCGTGGAAAGTCTTTAGGGTCTTCTACCTTTCGTATTCTTTTTAGGTGGGGTAGAGTGTTGGGATCCTATGCGTTTTTCATCATCACTGGCAAACATCTGATA3’) and BKPyV (5´TTCGTGGCTGAAGTATCTGAGACTTGGGCGTATGAGCATTGTGATTGGGATTCAGTGCTTGATGCGTTCCATGTCCAGAGTCTTCAGTTTCCTGATA 3’). The synthetic standard curves for Betaherpesvirus ranging from 5 to 5x108 genome copies/µL for HCMV, HHV-6 and HHV-7 were described by Raposo and Colleagues [68]
Raposo JV, Alves ADR, Dos Santos da Silva A, Dos Santos DC, Melgaço JG, Moreira OC, Pinto MA, de Paula VS. Multiplex qPCR facilitates identification of betaherpesviruses in patients with acute liver failure of unknown etiology. BMC Infect Dis. 2019 Sep 4;19(1):773. doi: 10.1186/s12879-019-4309-4. PMID: 31484497; PMCID: PMC6727340.
It is not clear what the authors are stating in Lines 167-172 and Table 4. Are the authors simply showing the percentage of people in their study group that were infected with HCMV? “Mono-infection or co-infection” is a bit confusing. Maybe just state HCMV infected?
We agreed and the table 4 was suppressed. The information from this table was add in the text. Line 165-170.
Again, in lines 194-195, it might be clearer to simply state HCMV infection.
As suggested, this information was changed in text.
Minor Issues:
- Line 45 is missing a parenthesis following the abbreviation “HCMV”
- Line 73: Formatting of the Citation should be addressed
- Line 120, 140, 149: spelling of “statical” should be corrected to “statistical.”
- Line 140 and footnote for Table 2: did the authors perform a “Tukey” test, not a “Turkey” test?
All these suggestions were observed in text and changed.
Reviewer 2 Report
Comments and Suggestions for Authors
This paper seeks to characterise the viral profile in salivary glands taken from patients diagnosed with a range of lesion types. This was done via nucleic acid extraction from parafilm-embedded samples followed by quantitative PCR using HCMV, HHV-6, HHV-7, BKPyV and JCPyV-specific primers.
There is a lack of clarity in the scientific processes undertaken in this study. Here are the main examples:
1) It is unclear what the "samples" actually refer to. Were the starting material whole salivary glands or the lesions themselves?
2) In the abstract, it is stated there are 68 samples but in the remainder of the paper it is stated there are 67 samples.
3) It is unclear what was actually extracted from the samples, was it DNA or RNA or both? In the methods describing the multiplex qPCR for HCMV, HHV-6 and HHV-7, it is stated that real-time qPCR was performed using a one-step RT-PCR kit. However the reaction mixture uses a PCR Enzyme mix and not RT-PCR. It is very confusing what was actually performed - reverse transcriptase PCR or quantitative/real-time PCR or both? If it was reverse transcriptase PCR - why? As the study focuses on DNA viruses.
4) The synthetic standard curves for JCPyV and BKPyV are described but not for HCMV, HHV-6 or HHV-7. It is unclear what these sequences are? A single-stranded nucleic acid product like an oligo? How would this work for PCR as it would not accurately relate to the dsDNA viral samples.
5) The results presented in tables are difficult to interpret. A bar graph or similar would be much better to convey the data. It is also unclear what the numbers mean - for example in Table 2 - what are the units? Are these just the means from samples where the virus was detected?
6) The epidemiological data (e.g. Table 1) and descriptions don't fit with the viral focus of this study.
Overall, until the scientific details of this study are described more accurately and with more details it is difficult to draw convincing conclusions from the data shown. For example, the conclusion on L289-291 is overreaching. It is well established that HCMV infects the salivary glands and that the saliva is an important transmission route. As it is unclear in the study what the actual samples are, the data here only reiterates what is already known and well established - that HCMV and other viruses are detected in the salivary glands.
Comments on the Quality of English Language
Extensive editing of the manuscript is required. The article is riddled with grammar errors, some of which make it difficult for the reader to understand meaning (e.g. L279). Additionally, some sentences seem to be incomplete (e.g. 274).
Author Response
This paper seeks to characterise the viral profile in salivary glands taken from patients diagnosed with a range of lesion types. This was done via nucleic acid extraction from parafilm-embedded samples followed by quantitative PCR using HCMV, HHV-6, HHV-7, BKPyV and JCPyV-specific primers.
There is a lack of clarity in the scientific processes undertaken in this study. Here are the main examples:
We would like to thank the reviewers for their time and the insightful comments and suggestions, which have improved the manuscript substantially. We believe that the modifications introduced in the revised version of the manuscript address the points raised. The changes incorporated are detailed below and highlighted in the revised report.
(5´TTCGTGGAAAGTCTTTAGGGTCTTCTACCTTTCGTATTCTTTTTAGGTGGGGTAGAGTGTTGGGATCCTATGCGTTTTTCATCATCACTGGCAAACATCTGATA3’) and BKPyV (5´TTCGTGGCTGAAGTATCTGAGACTTGGGCGTATGAGCATTGTGATTGGGATTCAGTGCTTGATGCGTTCCATGTCCAGAGTCTTCAGTTTCCTGATA 3’). The synthetic standard curves for Betaherpesvirus ranging from 5 to 5x108 genome copies/µL for HCMV, HHV-6 and HHV-7 were described previously by Raposo and Colleagues [68].
The synthetic curve is a Ultramer DNA Oligos that hybridized with probe and both primers. Ultramer DNA Oligonucleotides are generated by proprietary synthesis methods that deliver high-quality oligos up to 200 bases. They are available single- or double‑stranded.
5) The results presented in tables are difficult to interpret. A bar graph or similar would be much better to convey the data. It is also unclear what the numbers mean - for example in Table 2 - what are the units? Are these just the means from samples where the virus was detected?
We agree that was difficult to interpret table 2. Now, the data are showed in a figure as suggested.
6) The epidemiological data (e.g. Table 1) and descriptions don't fit with the viral focus of this study.
Although the main goal of the study is not an epidemiological, the information in the table 1 is important to describe the study population. We decided to keep this table in the manuscript, once the data of Betaherpesvirus in patients with adenoma and carcinoma are scarce.
Overall, until the scientific details of this study are described more accurately and with more details it is difficult to draw convincing conclusions from the data shown. For example, the conclusion on L289-291 is overreaching. It is well established that HCMV infects the salivary glands and that the saliva is an important transmission route. As it is unclear in the study what the actual samples are, the data here only reiterates what is already known and well established - that HCMV and other viruses are detected in the salivary glands.
The conclusion has been changed in the text at lines 312-318.
Results herein recorded for HCMV detected in salivary gland lesion regions have confirmed that HCMV infects the salivary glands, and that the saliva is an important transmission route used by it. Furthermore, HCMV co-infection with HHV-6 and HHV-7 in the investigated neoplasms has suggested that the synergy of these viruses can induce viral replication and increase viral load. However, further studies should be carried out to establish the association between these viruses and whether their synergy can overactivate some oncogenic regulatory pathways and epidermal growth factors.
Round 2
Reviewer 2 Report
Comments and Suggestions for Authors
All the issues should have been addressed in the authors' cover letter itself. Otherwise it is unclear if/where the issue has been addressed.
For example, in my original report I asked
1) It is unclear what the "samples" actually refer to. Were the starting material whole salivary glands or the lesions themselves?
Where is this addressed? Is it the amendment in the Materials and Methods "Salivary gland lesion regions". What does this actually mean? The lesion or including the organ around the lesion? How much of the sample is lesion:non-lesion cells?
Also in my original report
3) It is unclear what was actually extracted from the samples, was it DNA or RNA or both? In the methods describing the multiplex qPCR for HCMV, HHV-6 and HHV-7, it is stated that real-time qPCR was performed using a one-step RT-PCR kit. However the reaction mixture uses a PCR Enzyme mix and not RT-PCR. It is very confusing what was actually performed - reverse transcriptase PCR or quantitative/real-time PCR or both? If it was reverse transcriptase PCR - why? As the study focuses on DNA viruses.
Was this addressed in the paragraph starting "The AgPAth-IDtm One-Step RT-PCR kit" in the Discussion? One, this is not where this should be addressed. And two, the explanation is unconvincing. How does detecting both DNA and RNA indicate active infection/reactivation? Your methods can not differentiate them. And because you can not differentiate then you can not compare between your samples because the same cycle score in your RT-PCR from two different samples does not necessarily mean the viral DNA was detected at the same levels due to the cDNA also present.
4) The synthetic standard curves for JCPyV and BKPyV are described but not for HCMV, HHV-6 or HHV-7. It is unclear what these sequences are? A single-stranded nucleic acid product like an oligo? How would this work for PCR as it would not accurately relate to the dsDNA viral samples.
Presumably this is the response from the Methods section: Synthetic curve is a Ultramer DNA Oligos hybridized with probe and two primers. 372 Ultramer DNA Oligonucleotides are generated through proprietary synthesis methods 373 capable of delivering high-quality oligos up to 200 bases.
Who's proprietary? A company (name?) or the author's
5) The results presented in tables are difficult to interpret. A bar graph or similar would be much better to convey the data. It is also unclear what the numbers mean - for example in Table 2 - what are the units? Are these just the means from samples where the virus was detected?
There is no label for the y-axis and therefore I still don't know what the unit is.
The comments and suggestions I gave in my first report have not been adequately addressed.
Comments on the Quality of English Language
I asked for extensive editing of the language giving two examples. I presume the two examples have been addressed but I do not know where they are in the revised manuscript and have not been given any signposts of where they are. There is no evidence of any other language editing elsewhere.
Author Response
Reviewer 2:
1) It is unclear what the "samples" actually refer to. Were the starting material whole salivary glands or the lesions themselves?
Where is this addressed? Is it the amendment in the Materials and Methods "Salivary gland lesion regions". What does this actually mean? The lesion or including the organ around the lesion? How much of the sample is lesion:non-lesion cells?
In this study the samples collected were lesion including the salivary gland. All samples had lesion cells. This information was addressed in line 319-320 in Material and Methods.
3) It is unclear what was actually extracted from the samples, was it DNA or RNA or both? In the methods describing the multiplex qPCR for HCMV, HHV-6 and HHV-7, it is stated that real-time qPCR was performed using a one-step RT-PCR kit. However the reaction mixture uses a PCR Enzyme mix and not RT-PCR. It is very confusing what was actually performed - reverse transcriptase PCR or quantitative/real-time PCR or both? If it was reverse transcriptase PCR - why? As the study focuses on DNA viruses.
The kit MagMax FFPE DNA/RNA Ultra Kit (Thermo Fisher Scientific, Waltham, MA, USA can extract DNA and RNA. We performed the quantitative/real-time PCR focuses on DNA detection.
The name of the kit was wrong. In this study we used the kit Taq-Man™ Universal PCR Master Mix (Thermo Fisher Scientific, Waltham, MA, USA). We apologized for the mistake. This information was fix in the manuscript in lines 343-350.
Was this addressed in the paragraph starting "The AgPAth-IDtm One-Step RT-PCR kit" in the Discussion? One, this is not where this should be addressed. And two, the explanation is unconvincing. How does detecting both DNA and RNA indicate active infection/reactivation? Your methods can not differentiate them. And because you can not differentiate then you can not compare between your samples because the same cycle score in your RT-PCR from two different samples does not necessarily mean the viral DNA was detected at the same levels due to the cDNA also present.
The explanation of AgPAth was deleted in the discussion. In this study the DNA were detected by quantitative/ real time PCR. The kit used Taq-Man™ Universal PCR Master Mix (Thermo Fisher Scientific, Waltham, MA, USA).
You are right and we completely agree that the methods can not differentiate DNA and RNA and the only detection of DNA not distingue infection and reactivation. The viral loads of DNA and RNA were not compared in our study.
The methods were rewritten with the right kit.
Line 343-350: Quantitative/Real-time (qPCR) was performed, based on using commercial Taq-Man™ Universal PCR Master Mix (Thermo Fisher Scientific, Waltham, MA, USA), in order to confirm viral detection, as well as to measure viral load through HCMV, HHV-6 and HHV-7 target regions U54, U56 and U37, respectively. However, this assay could not differentiate HHV-6A from HHV-6B in the analyzed samples. Multiplex qPCR was performed according to manufacturer's instruction; the reaction mixture comprising 1 μL 25x PCR Enzyme, 1 μL of each oligonucleotide (3 μM), 1 μL probe (0.4 μM), 12.5 μL of 1x PCR Buffer and 2,5 μL DNA.
4) The synthetic standard curves for JCPyV and BKPyV are described but not for HCMV, HHV-6 or HHV-7. It is unclear what these sequences are? A single-stranded nucleic acid product like an oligo? How would this work for PCR as it would not accurately relate to the dsDNA viral samples.
The synthetic standard curves for HCMV, HHV-6 or HHV-7 were described previously in the manuscript published by Raposo et al., 2019. The manuscript was referred in this study.
Lines 384-392: Synthetic standard curves for Betaherpesvirus, (HCMV 5’-TTCGTGGCCTCGTAGTGAAAATTAA TGGTCGTATTTGAACAGATCGCGCA CCAATACGGATGCGTTCCTGCAGA CAGTAACGGCCCTGATA-3’, HHV-6 5’-TTCGTGCAAGCTCATGAACATCGTCACGTATACCGATCCCAGCTCACCACCATCTAAATGCGTAGGTAGCG GCAATTTAGGTCTTTCTGATA-3’ and HHV-7 5’-TTCGTCCAATCCTTCCGAA ACCGATCGTATCATGGCCAACAAGCAATCTGCGAGATGCGTTTGTCATTACTCCAGTGA CTTCCGCTGATA-3’) ranging from 5 to 5x108 genome copies/µL of HCMV, HHV-6 and HHV-7, this sequences were previously described by Raposo and Colleagues [71].
The synthetic curves by Integrated DNA Technologies, Inc., (IDT) work as single strand. It is based on the three steps of PCR required for any DNA synthesis reaction. The first step is the denaturation of the template into single strands. Here the standard curve is the single strand that contain the sequences that hybridize with primers and probes. The ultramer DNA are designed as primer reverse in the same direction, probe and primer forward as reserve complement.
Presumably this is the response from the Methods section: Synthetic curve is a Ultramer DNA Oligos hybridized with probe and two primers. Ultramer DNA Oligonucleotides are generated through proprietary synthesis methods capable of delivering high-quality oligos up to 200 bases.
Who's proprietary? A company (name?) or the author's
Lines 374-375: The company name is Integrated DNA Technologies, Inc., (IDT) Coralville, Iowa, USA.
5) The results presented in tables are difficult to interpret. A bar graph or similar would be much better to convey the data. It is also unclear what the numbers mean - for example in Table 2 - what are the units? Are these just the means from samples where the virus was detected? There is no label for the y-axis and therefore I still don't know what the unit is.
As suggested a bar graph was added in the manuscript. We apologized, the label for y-axis was informed in the figure 1 “DNA viral load (copies/mg)”
The comments and suggestions I gave in my first report have not been adequately addressed.
We hope that this answers adequately addressed the questions that were not answered in the previous rebuttal letter.
